# Phenotyping of Potato Plants Using Morphological and Physiological Tools

**DOI:** 10.3390/plants13050647

**Published:** 2024-02-26

**Authors:** Olga Rozentsvet, Elena Bogdanova, Viktor Nesterov, Alexey Bakunov, Alexey Milekhin, Sergei Rubtsov, Victor Rozentsvet

**Affiliations:** 1Samara Federal Research Scientific Center RAS, Institute of Ecology of the Volga Basin RAS, 10, Komzina, Togliatti 445003, Russia; cornales@mail.ru (E.B.); nesvik1@mail.ru (V.N.); rozentsvet@mail.ru (V.R.); 2Samara Federal Research Scientific Center RAS, Samara Scientific Research Agriculture Institute Named after N.M. Tulaykov, Bezenchuk 446254, Russia; bac24@yandex.ru (A.B.); alekseimilehin@mail.ru (A.M.); rubcov_sl@mail.ru (S.R.)

**Keywords:** *Solanum tuberosum*, morphometric and morpho-physiological traits of leaves, phenotyping

## Abstract

Potato (*Solanum tuberosum* L.) is one of the main non-grain agricultural crops and one of the main sources of food for humanity. Currently, growing potatoes requires new approaches and methods for cultivation and breeding. Phenotyping is one of the important tools for assessing the characteristics of a potato variety. In this work, 29 potato varieties of different ripeness groups were studied. Linear leaf dimensions, leaf mass area, number of stems, number of tubers per plant, average tuber weight, signs of virus infection, dry weight, pigment content, and number of stomata per unit leaf area were used as phenotyping tools. The strongest positive relationship was found between yield and bush area in the stage of full shoots (R = 0.77, *p* = 0.001), linear dimensions of a complex leaf (R = 0.44, *p* = 0.002; R = 0.40, *p* = 0.003), number of stems (R = 0.36, *p* = 0.05), and resistance to viruses X (R = 0.42, *p* = 0.03) and S (R = 0.43, *p* = 0.02). An inverse relationship was found between growth dynamics and yield (R = −0.29, *p* = 0.05). Thus, the use of morphological and physiological phenotyping tools in the field is informative for predicting key agricultural characteristics such as yield and/or stress resistance.

## 1. Introduction

Observed climate changes, world population growth, and agricultural land area decreases require accelerated research activities on increasing the yield potential of cultivated plants as well as their adaptation to stressful conditions [1,2]. The time-efficient and precise phenotypic assessment of thousands of breeding lines, clones, or populations over time under different environmental conditions is a kind of faster development of new and improved varieties [3,4,5]. This approach can predict some complex traits that are important for plant breeding (forward phenomics) and also provide explanations for why given genotypes are bred in a particular environment (reverse phenomics) [6,7].

A phenome is a set of phenotypes, which are defined as organism characteristics arising as a result of interactions between genotype, environment, and agricultural technologies [5,8,9]. A feature of plant phenomics is the necessity to take into account the wide range of conditions (temperature, humidity, light, soil type, etc.) since it is assumed that up to 50% of phenotypic variations can be caused by environmental factors [10]. One of the most important phenotype characteristics is its multilevel nature as far as the manifestation of the genome can be described at all levels of the living system organization, from the molecular level to the whole organism [5,11].

The region of interest of breeders focuses on the study of phenotypic characteristics at the level of organs such as the roots, leaves, stems, inflorescences, ears, and tubers; plant physiological properties (development rate in individual stages of ontogenesis, photosynthesis efficiency indicators and water use efficiency, sustainability to stress); or their general characteristics such as productivity, biomass, and disease resistance [12,13]. Many breeding programs focus on commercially important traits. It was shown that plant productivity along with economic yield was closely related to characteristics associated with plant growth and development [14]. For example, a relationship between leaf area and biomass accumulation in *Arabidopsis thaliana* was established [15]. Our preliminary work examined the stability and plasticity of potato yields of different ripeness groups depending on the environmental conditions [16,17].

In these latter years, plant phenotyping technologies have become widespread based on the study of images and the calculation of phenotypes through the accurate analysis of a large number of plants during the short period of time [9,18,19]. This involves the use of an automated equipment complex for non-contact analysis of plants [20,21].

Despite the outstanding success of high-throughput phenotyping, it should be noted that most breeding research has been developed for model plants and cereals [22]. Work on other groups of species is carried out discretely (for example, fruit trees, forage crops, flowers, or forestry) [3,7]. Most phenotypic assessments are being performed under controlled conditions [23,24]. However, the results do not correspond well to field trials [25]. In addition, when conducting large-scale phenotyping experiments, one of the bottlenecks is the search and measurement of the most informative traits that are important for identifying productive and resistant genotypes. It should be noted that the assessment of many quantitative traits can be carried out using traditional classical methods that do not require the use of expensive phenotyping platforms, especially in the stage of preliminary selection and in small agrocenoses [20,26].

Potato (*Solanum tuberosum* L.) is one of the main non-grain agricultural crops and one of the main sources of food for humanity [27]. A biological feature of this crop is its vegetative propagation. With this type of reproduction, new offspring plants come the individual vegetative organs of the mother plant, a change in generations does not occur, and genetically homogeneous groups of individuals are formed [28].

The main economically useful trait of potatoes is the tuber yield [25]. The formation of the crop is carried out during the production process that is a complex and integrated function of plants. It is based on the genetically determined processes of growth and development. Potato plants are usually classified depending on their ripening time. The optimal selection of varieties for each specific region is one of the main factors influencing the increase in productivity and quality of food and seed potatoes [17].

As the storage organs of the potato plant, tubers are formed due to the accumulation of photosynthesis products on the tops of the stolons. The size of the leaf surface determines the activity of absorption of solar radiation and is the main factor on which the yield depends [29]. Lack of moisture and high temperatures lead to a decrease in the growth and development of plants, the formation of new leaves decelerates, and plant aging gathers pace, especially after flowering [30,31]. The formation of tubers is inhibited as a result, which leads to a limited number and weight of tubers [32].

Potatoes are among the crops that are severely affected by viral pathogens, which pose a particular danger [33]. Symptoms of a viral infection often include the yellowing or mosaicking of infected leaves, the appearance of pigmentation that is determined by structural and functional damage, and changes in leaf cell homeostasis [34]. Therefore, leaf parameters are a valuable trait in assessing potato yield.

For phenotyping potato plants, criteria such as the size and condition of the above-ground mass [25], traits associated with photosynthesis [35], and quantitative assessment of the color and shape of tubers [36] are used. However, the need for phenotypic assessment of genotype–phenotype interactions in breeding programs is not decreasing [37,38,39].

The aim of this research was to study the phenotypic variability of potato genotypes using visual morphological and physiological traits associated with yield. The tasks of this study were (i) the assessment of the yield of 29 new potato varieties in different varietal groups grown in the climatic conditions of the Middle Volga region of Russia in 2022–2023; (ii) the study of growth dynamics according to phenological indicators; and (iii) the test of resistance to viral diseases.

## 2. Results

### 2.1. Climate Condition, Planting, Growing, and Harvesting Potatoes

Potato growing conditions in different years of this study varied by temperature regime and moisture availability. Average air temperatures in certain periods of 2023 were higher in comparison with 2022. In particular, air temperatures in May—the period of planting and the appearance and germination of seedlings—were higher on average by 7.4 °C, and in August—the period of tuber ripening—by 1.4 °C. In May and July 2023, a higher level of maximum air temperatures was also noted, namely 31.6 °C in May versus 23.8 °C, and also in July—37.3 °C versus 32.9 °C. The amount of precipitation in 2023 was also significantly lower for the entire growing season, but especially in May, when the level of precipitation for both years was 4.7 times lower compared to 2022 (Table 1).

Fluctuations in temperature and moisture conditions affected the yield parameters of potato varieties. The total yield of early-ripening potato varieties over two years was 26.1 t/ha (Figure 1A).

Potato varieties with middle-early periods demonstrated higher yields—34.2 and 32.5 t/ha. At the same time, significant changes in yield components were discovered. In particular, there was a decrease in the number of tubers per plant in 2023, especially in the group of early- and mid-season-ripening varieties (Figure 1B). At the same time, the average tuber weight in varieties of the early and middle-early groups increased, while in the mid-season-ripening group, it declined (Figure 1B).

Consequently, the main differences in the yield of potato in 2023 from the previous year of reproduction are associated with a decrease in the overall yield of mid-season varieties due to a decrease in the number and weight of tubers. In early and mid-early varieties, the number of tubers decreased, but their weight increased. At the same time, varieties were identified, the average yield of which over two years was comparable to or exceeded the yield of the standards (Appendix A). Among the early varieties (Gala standard) were Sinichka, Kholmogorsky, Argo, Blossom, and Krasnoyarskyranniy. Among the middle-early ones (Ilyinsky standard) were Ariel, Kaluzhsky, and Alka. Among the mid-season-ripening varieties (Zhigulevsky standard) was Tarasov only. However, judging by the dynamics of the yield of these varieties, only four varieties (Blossom, Krasnoyarskyranniy, Dvinsky, and Kaluzhsky) had a higher yield in the second year than in the first year of seeding. The yield variability coefficients for these varieties did not exceed 20%. For the variety Alka, the average yield for two years was 44.6 t/ha, which was higher in comparison with the average yield of the group of middle-early varieties by 19% (Cv = 16.8%). The same variety was distinguished by an optimal, fairly constant ratio of the number of tubers and tuber weight per plant.

### 2.2. Duration of Phenological Phases for Potato Plants

The general trend of changes in the duration of phenological phases depending on the timing of potato ripening in different years is shown in Figure 2A,B.

The duration of the period from planting to full germination was longer in the first year of this study compared to the second year. It decreased in early and middle-early varieties. The period “full germination-beginning of flowering” was greatest in the second year in the group of middle-early varieties. The duration of flowering was greatest in mid-season-ripening varieties in the first year of cultivation and practically did not differ between groups in the second year.

More detailed analysis of phenological phases showed that the phase from planting to full germination in early and middle-early varieties in the first year of reproduction was 22–26 days (Appendix A). In the second year, it varied significantly within each group, from 18 to 28 days. Moreover, the coefficient of variation in the group of early ripening varieties varied from 3.1 to 25.7%, while for later varieties, this interval was 0–28%. The range of variation in the second and third phenological phases had higher values compared to the first phase (0–40% and 0–79%), which indicates a greater dependence of the flowering duration on the potato genotype and its response to environmental conditions.

### 2.3. Visual Recording of Viral Diseases of Potato Leaves

The analysis of visual recording of viral diseases on potato plants showed that the group of early-ripening varieties is more susceptible to potato virus X compared to varieties of the middle-early and mid-season-ripening groups (Figure 3A–D). Thus, the indicator of resistance to virus X in the group of early-ripening varieties decreases from the value of 7.9 to 7.7. An even greater decrease in resistance in all groups of varieties was noted for potato viruses Y and S. Moreover, all varieties exhibited high resistance to L (potato leafroll virus). Varieties with high field resistance to the most harmful viruses Y and X have been identified as the following: Krasnoyarskyranniy (eary-ripening), Orlan (middle-early), and Armada and Evpatiy (mid-season-ripening). These varieties can be characterized as the most resistant to viral diseases (Appendix A). 

### 2.4. Morphometric Traits of Potato Varieties

Phenotypic variability was evaluated by the ground mass growth rate and morphometric parameters of the whole plant and individual leaves in potato varieties with different ripening periods (Figure 4). To exclude annual fluctuations in the measured traits, average values were calculated for two years.

The growth of the above-ground parts of plants continues until the full flowering phase and is largely determined by the timing of the ripening of the variety (Figure 4A). The height of plants after the emergence of full shoots (30 days) was greatest in the group of early-ripening varieties but became smaller by the period of full flowering in comparison with the varieties of other ripening groups. The highest growth rate of all studied varieties was in the first 30–40 days (Figure 4B). In mid-season-ripening varieties, growth was faster than in varieties with earlier ripening periods throughout the entire growing season but especially in the first 30–40 days (in the stage from full germination to the beginning of flowering). Further, the growth rate decelerated and became minimal by the full flowering phase.

When measuring morphometric parameters, it was found that varieties of the middle-early ripeness group had a larger number of stems per plant and a total larger leaf surface area (Figure 5A,B) as well as larger lengths of the lateral lobes of the leaves and the complex leaf as a whole (Figure 5C,D).

### 2.5. Morpho-Physiological Traits

The duration of phenological phases is closely related to growth and physiological processes. The leaves of plants of earlier ripening periods were denser compared to the leaves of later varieties, the specific LMA of which was 1.2–1.4 times lower (Figure 6A).

The number of stomata that provide CO_2_ input and regulate transpiration flow varied little on average between groups (Figure 6B). The total photosynthetic pigment content in the leaves of middle-early varieties was only 10–15% higher compared to early ripening and mid-season-ripening varieties (Figure 6C). Differences in leaf dry matter content were not significant (Figure 6D).

The data obtained made it possible to identify intragroup and intergroup differences in genotypes in growth rate, resistance to viral diseases, yield, as well as morphometric and physiological parameters.

### 2.6. Relationship between Morphometric and Physiological Parameters of Leaves and Yield

A comparison of the studied varieties showed that within each of the groups, differentiated by ripening time, there are varieties with high and low yields, with the exception of the group of mid-early varieties (Figure 7A). We divided all varieties into three groups. The first group included low-yielding varieties (LYV). The second and third groups included standard varieties (SV) and high-yielding varieties (HYV) (F = 38.1, *p* = 0.0001) (Figure 7A), respectively.

Using discriminant analysis, we obtained clear differences between groups in terms of the totality of morphological and physiological characteristics (Figure 7B).

The yield of high-yielding varieties was 1.5–2.0 times higher than that of low-yielding varieties and standards (Figure 8A,B). The group of high-yielding varieties was distinguished by a faster transition from planting tubers to the appearance of full-fledged seedlings and greater plant height (Figure 8D,E). The largest number of stems was found in standard varieties (Figure 8F). At the same time, in high-yielding varieties, the length and width of the complex leaf were 1.1 and 1.2 times greater compared to standards and low-yielding varieties, which generally created a large photosynthetic surface as a guarantee of photosynthesis efficiency (Figure 8A,D,H). Larger areas of bushes in high-yielding varieties were observed during the period of full germination 30–40 days after planting (Figure 8I). It was also revealed that the field resistance of a group of high-yielding varieties to potato viruses X, Y, and S was higher than those of standards and low-yielding varieties (Figure 8K,L). However, the characteristics, such as LMA and number of stomata, differed little among the selected groups.

Statistical scatterplots of morphological and physiological parameters were used to identify genotype–phenotype relationships. Figure 9 shows data that were most correlated with the yield of 29 potato varieties. The strongest positive relationship was found between potato yield (*X*-axis) and bush area on the 30–40th day of development (*Y*-axis) (R = 0.77, *p* = 0.001) (Figure 9A–F). During the flowering period, the linear dimensions of the complex leaf (R = 0.44, *p* = 0.002; R = 0.40, *p* = 0.003) and the number of stems (R = 0.36, *p* = 0.05) correlated with the yield. Resistance to viral infections also had a positive effect on the yield of virus X (R = 0.42, *p* = 0.03) and virus S (R = 0.43, *p* = 0.02). An inverse relationship was found in relation to growth dynamics and yield (R = −0.29, *p* = 0.05).

The data obtained show that the architecture of the above-ground mass as a whole is an important parameter in the formation of potato yield.

## 3. Discussion

Potatoes are grown in more than 100 countries around the world as an important foodstuff due to their ability to adapt to a wide range of environmental conditions [2,19]. However, growing potatoes as well as many other agricultural crops requires new approaches and methods for cultivation and breeding [40]. Plant phenotyping is one of the new approaches to assessing the physiological characteristics of a species, variety, or line [3,8,19]. For example, the use of the leaf area index in monitoring the growth of potato plants has been demonstrated [25]. Another example is the use of phenotyping based on root and shoot characteristics to identify yield differences between potato genotypes [38].

Crop yield is an important indicator in breeding programs [17]. As a result of two-year field experiments, we identified significant variability in the yield of 29 new potato varieties within varietal groups differentiated by ripening time. Moreover, in each group, varieties were found that differed in yield from the standards within the varietal group and between groups. The group of middle-early varieties is the most preferable group in terms of overall yield and its stability in the conditions of the Middle Volga region where the largest number of high-yielding varieties was found, in which the tubers had, on average, a large mass and their number decreased to a lesser extent compared to other varietal groups.

Monitoring crop growth and productivity during development is an important aspect of phenotyping [25]. The main increase in above-ground biomass and the formation of potato tubers in the conditions of the Middle Volga region occurred in June and July. Under the influence of higher temperatures in the third ten days of May 2023, plants of the early and middle-early group germinated faster compared to 2022, and higher maximum temperature values in July caused a decrease in the duration of the full flowering period in all varietal groups. At the same time, the leaf surface growth rate changed, which was one of the important prerequisites for the accumulation of economically useful biomass and the formation of the crop [29]. It is known that unfavorable water and temperature conditions primarily contribute to a decrease in the number of tubers (40). In our studies, the negative influence of the environmental conditions also had a greater effect on the reduction in the number of tubers than on their weight (Figure 1).

Phenotyping of plants can be performed based on several morphological, physiological, biochemical, and molecular factors [6,12]. Characteristics associated with the growth and development of potato plants are plant architecture, leaf structure, etc. [14,38]. As phenotyping tools, we used a set of parameters measured by traditional methods in combination with imaging methods (see Materials and Methods). A comparison of these characteristics between different ripeness groups shows that the varieties of the middle-early group were characterized by a greater height and area of the bush, larger leaf sizes, and a greater number of stems (Figure 5). To get a clearer picture of the relationship between the studied traits and yield, we divided the plants into groups with different yields. It turned out that the group of high-yielding varieties, along with larger values of height, leaf size, and bush area, were distinguished by later dates of emergence of seedlings and a smaller number of stems per plant in comparison with the standards (Figure 8). An important trait characterizing the architecture of the bush turned out to be the ratio of the leaf mass area to the height. The number of stems can have different effects on overall potato yield. This is because the number of tubers produced depends on competition between stems for growth factors such as nutrients, water, and light. At lower stem numbers, there is less competition, resulting in more tubers per stem [12,13,31]. Finally, with an increase in the number of plants affected by viral diseases in the second year of reproduction, resistance in the group of high-yielding varieties was generally higher compared to other groups.

An analysis of variance allowed us to identify a potential relationship between genotype and phenotype. The closest relationship between the yield of the genotype was revealed with such characteristics of phenotypes as large plant heights and total leaf area, and the ratio of bush height to leaf mass area, but at a slower growth rate. The highest correlation coefficient between bush area and area-to-height ratio suggests that this is an important trait that breeders should pay attention to.

## 4. Materials and Methods

The objects of study were 29 potato varieties of different ripeness groups and genetic origins. Early-ripening varieties—Gala (standard), Sprinter, Sinichka, Kholmogorsky, Polyarny, Bashkirsky, Argo, Blossom, Krasnoyarskyranniy; middle-early—Ilyinsky (standard), Ariel, Dvinsky, Kaluzhsky, Bagira, Orlan, Alka, Visa, Farn; mid-season-ripening—Zhigulevsky (standard), Evpatiy, PrincessaNatavan, Tomichka, Chaika, Tersky, Intelligent, Spiridon, Tarasov, Moryak, Dalnevostochny.

Field research was carried out on the experimental plot of the Samara Scientific Research Agriculture Institute named after N.M. Tulajkov—branch of the Samara Federal Research Scientific Center RAS in 2022–2023 (Middle Volga region, Russia; 53°03′ N, 49°25′ E). Potato tubers were planted in the second year of May in four replicates of 50 tubers each and were grown without irrigation. Soils were terraced chernozem, ordinary, low-humus, medium-thick, heavy loamy. Harvesting was carried out simultaneously for all potato varieties at the end of August.

The influence of meteorological conditions on plant vegetation and crop yields over two studying years was assessed using data from the average daily and maximum air temperature and the precipitation amount. Weather data were provided by the hydrometeorological station of the city of Bezenchuk, Samara region.

Phenological stages of potato growth included the duration of the periods from the beginning of planting tubers to the appearance of the first shoots, from full germination to the beginning of flowering, and the flowering.

The potato productivity of each variety was assessed by the total yield (t/ha), the number of tubers per plant (pcs.), and the average weight of one tuber (g).

The field resistance of potato varieties and hybrids to viral diseases was assessed visually twice during the periods of budding and flowering against a natural infectious background on the following scale: 9 points—very high resistance (0–10% of affected plants); 7 points—high resistance (11–25% of affected plants); 5 points—average resistance (26–50% of affected plants); 3 points—low resistance (5–75% of affected plants); 1 point—very low resistance (75–100% of affected plants, plant death).

Plant height was measured proximally in 15–20 plants using a ruler. Growth dynamics were determined by plant height on the 30th, 40th, and 50th days and are expressed as %.

We used the method of photographing plants on the 30–40th days using a NikonD-3200 digital camera (Nikon Corporation, Tokyo, Japan) to work with the phenotype. To determine the area of a plant bush, 320 photographs were taken from the side at right angles and from above to minimize errors that could arise during photo processing.

Images of leaves were obtained from 15–20 plants, 3–4 leaves each, during the flowering period of the plants. All received photographs were processed in the JMicroVision-v.1.3.4 program (Geneva, Switzerland).

Specific leaf area (SLA) was determined by obtaining cuttings from the middle part of 5–10 leaves for each variety which were then weighed, dried to a constant weight, and used for calculation.

The specific LMA was determined by the formula *m*/*S* (mg/dm^2^), where *m* is the amount of dry mass (mg), *S* is the leaf surface area (dm^2^).

Stomata were counted on longitudinal paradermal sections of leaves, previously fixed in 3.5% glutaraldehyde (pH = 7.5). The number of stomata per 1 cm^2^ on the lower surface of the leaf was counted and their length was measured, expressed in µm. A biological microscope “OPTIKA” B-500TPL (Ponteranica, Italy) was used for analysis.

To measure physiological parameters, the lateral lobes of a compound leaf of the middle tier were collected from 15–20 plants of each variety. The leaves were cut and mixed, and three biological samples of 0.1–0.5 g each were taken from the combined mass, depending on the type of analysis. Each biological sample was analyzed three times.

Photosynthetic pigments were extracted with cold acetone (90%) from 0.5 g of fresh frozen leaves. The content of pigments was determined in the resulting acetone extract using a PromEcoLab PE-3000 UV spectrophotometer (Shanghai Mapada Instruments Co., Ltd., Shanghai, China). The concentrations of chlorophylls a and b were measured at wavelengths of 662 and 645 nm, respectively, and carotenoids were measured at 470 nm. The concentration of isolated pigments was calculated according to the method in [41].

The leaf dry weight was determined after drying them to constant weight at a temperature of 60 °C and is expressed as a percentage of wet weight.

Statistics. Data in tables are presented as absolute values: as average values for a group of varieties in bar graphs and as average values of measured parameters for each variety in scatter plots. One-way ANOVA test was used with the Bonferroni correction to prove significant differences between the average values of morpho-physiological parameters with normal data distribution, and in cases of deviation from the normal distribution, with the Kruskal–Wallis criteria. To establish differences between isolated groups of potato plants according to environmental or economic (yield) characteristics, discriminant analysis was used. Calculations were performed using Statistica 6.0 for Windows, Past 3, Statgraphics Centurion XVI, and Microsoft Excel 2007 software.

## 5. Conclusions

The results of this study allowed us to draw the following conclusions. For the effective use and development of new varieties, it is extremely important to take into account the climatic conditions of the region in which a certain potato variety will be cultivated. Selecting the optimal variety can help make important decisions such as determining the optimal time to harvest based on varietal variation.

In the pre-breeding stage, inexpensive visual methods for determining phenotypes of potatoes are informative. There should be a focus on phenotypic characteristics associated with yield such as plant height, total leaf area, the optimal ratio of bush height to leaf mass area in the stage of full seedling emergence, and the duration of the period from the stage of planting tubers to the stage of full germination when choosing a promising variety for practical use or as a source of genetic material.

## Figures and Tables

**Figure 1 plants-13-00647-f001:**
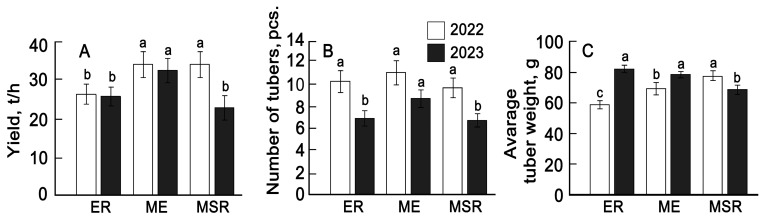
The average values of yield (**A**), number of tubers (**B**), and tuber weight (**C**) of 29 potato varieties for 2022–2023. ER—early ripening (*n* = 9); ME—middle-early (*n* = 9); MSR—mid-season ripening (*n* = 11). Each bar represents the mean ± SE (*n*—number of varieties). Different letters indicate statistically significant differences (one-way ANOVA with the Bonferroni correction, *p* < 0.05).

**Figure 2 plants-13-00647-f002:**
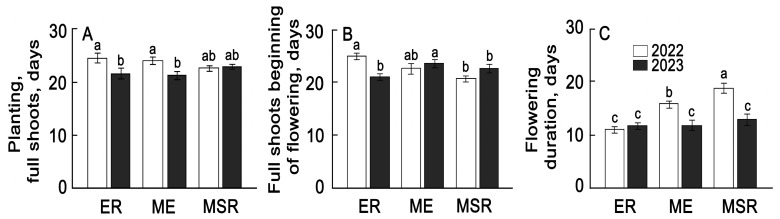
Average indicators of the duration of phenological phases for 29 potato varieties plants for 2022–2023. Duration of the period: planting tubers—shoots (**A**); shoots—beginning of flowering (**B**); potato flowering (**C**). ER—early ripening (*n* = 9); ME—middle-early (*n* = 9); MSR—mid-season ripening (*n* = 11). Each bar represents the mean ± SE (n—number of varieties). Different letters indicate statistically significant differences (one-way ANOVA with the Bonferroni correction, *p* < 0.05).

**Figure 3 plants-13-00647-f003:**
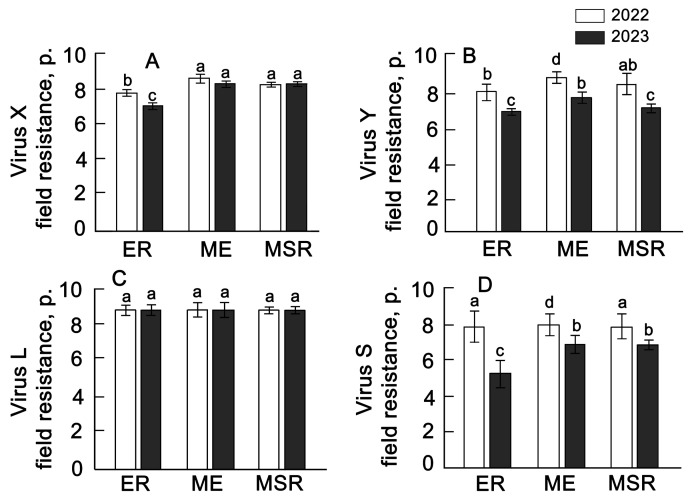
Field resistance of 29 potato varieties plants for 2022–2023 to the X virus (**A**), Y virus (**B**), L virus (**C**), S virus (**D**). ER—early ripening (*n* = 9); ME—middle-early (*n* = 9); MSR—mid-season ripening (*n* = 11). Each bar represents the mean ± SE (*n*—number of varieties). Different letters indicate statistically significant differences (one-way ANOVA with the Bonferroni correction, *p* < 0.05).

**Figure 4 plants-13-00647-f004:**
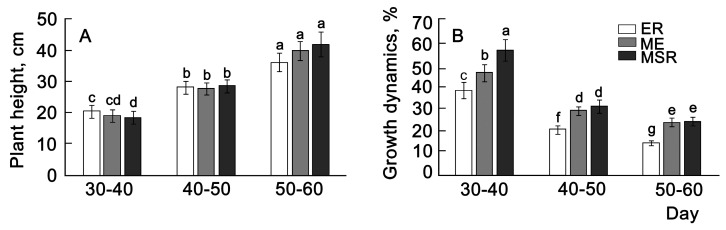
Dynamics of average values of height (**A**) and growth (**B**) of plants of 29 potato varieties over two years. ER—early ripening (*n* = 9); ME—middle-early (*n* = 9); MSR—mid–season ripening (*n* = 11). Each bar represents the mean ± SE (*n*—number of varieties). Different letters indicate statistically significant differences (one-way ANOVA with the Bonferroni correction, *p* < 0.05).

**Figure 5 plants-13-00647-f005:**
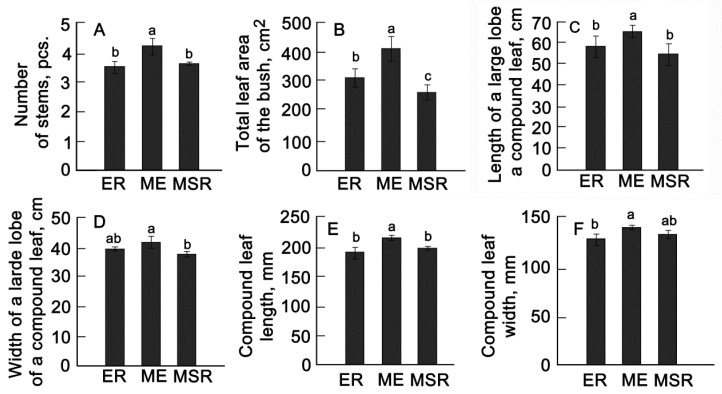
Average values of morphometric traits of potatoes of different varieties: the number of stems (**A**), the total area of the bush (**B**), the length of a large lobe of a compound leaf (**C**), the width of a large lobe of a compound leaf (**D**), the length of a compound leaf (**E**), the width of a compound leaf (**F**). ER—early ripening (*n* = 9); ME—middle-early (*n* = 9); MSR—mid–season ripening (*n* = 11). Each bar represents the mean ± SE (*n*—number of varieties). Different letters indicate statistically significant differences (one-way ANOVA with the Bonferroni correction, *p* < 0.05).

**Figure 6 plants-13-00647-f006:**
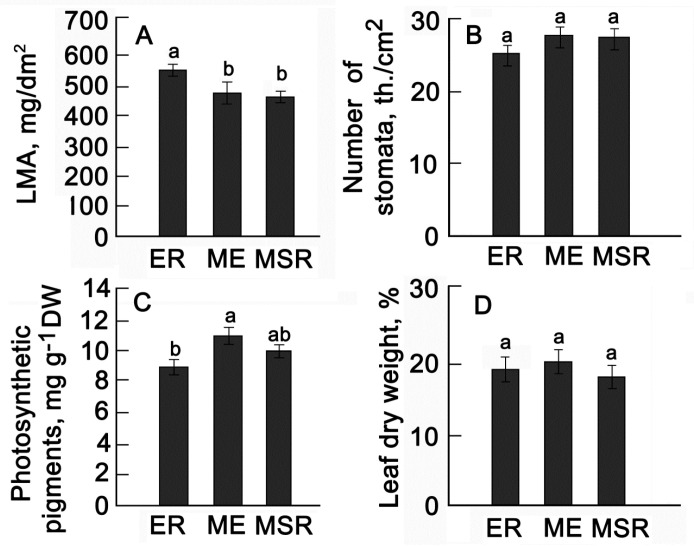
Average values of morpho-physiological traits: leaf surface area (**A**), number of stomata (**B**), photosynthetic pigments (**C**), leaf dry weight (**D**). ER—early ripening (*n* = 9); ME—middle-early (*n* = 9); MSR—mid–season ripening (*n* = 11). Each bar represents the mean ± SE (*n*—number of varieties). Different letters indicate statistically significant differences (one-way ANOVA with the Bonferroni correction, *p* < 0.05).

**Figure 7 plants-13-00647-f007:**
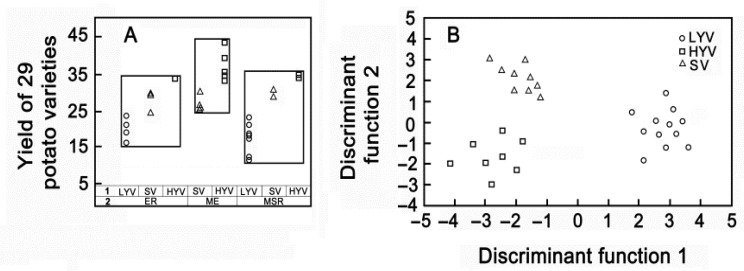
Variability plot (**A**) and Ordination diagram for discriminant analysis (**B**) using the *p* < 0.05 chi-square subset; 29 cases were used to develop a model to discriminate among the 3 groups of potato varieties. Designations: (1) HYV—high-yielding varieties (*n* = 8); SV—standards and varieties equal to standards (*n* = 9); LYV—low-yielding varieties (*n* = 12) (*n*—number of varieties); (2) ER—early ripening (*n* = 9); ME—middle-early (*n* = 9); MSR—mid-season ripening (*n* = 11).

**Figure 8 plants-13-00647-f008:**
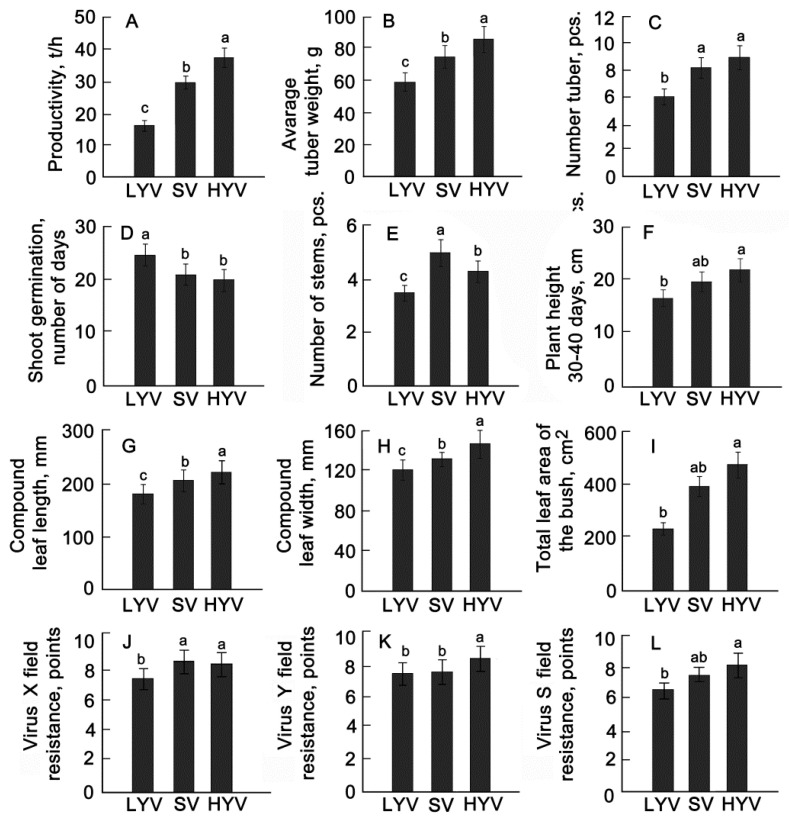
Elements of potato productivity (**A**–**C**), morpho-physiological characteristics of leaves (**D**–**I**), and field virus resistance (**J**–**L**) of different groups of potato. HYV—high-yielding varieties (*n* = 8); SV—standards and varieties equal to standards (*n* = 9); LYV—low-yielding varieties (*n* = 12) (*n*—number of varieties). (**A**)—Resistance to virus X; (**B**)—resistance to virus Y; resistance to virus S (**C**). Different letters indicate statistically significant differences (one-way ANOVA with the Bonferroni correction, *p* < 0.05).

**Figure 9 plants-13-00647-f009:**
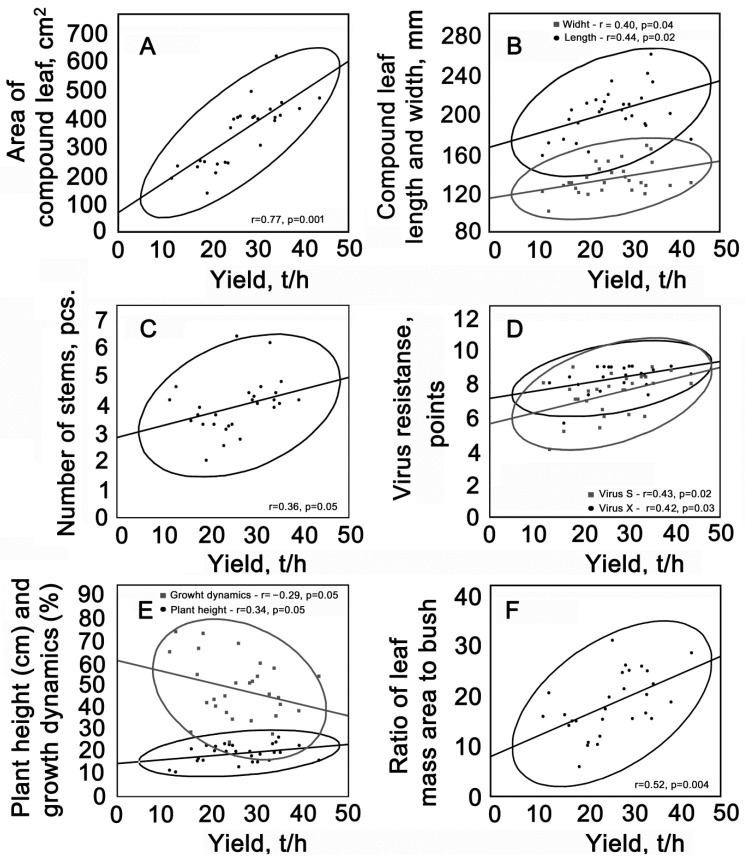
Scatter plots showing the relationship between: area of compound leaf (**A**), compound leaf length and width (**B**), number of stems (**C**), virus resistanse (**D**), plant height and growth dynamics (**E**), ratio of leaf mass area to bush (**F**) and the yield of 29 potato varieties.

**Table 1 plants-13-00647-t001:** Climatic conditions of the growing season 2022–2023.

Month	Ten Days of the Month	Average Daily Temperature, °C		Maximum Temperature, °C		Precipitation, mm		Growth Stage
		2022	2023	2022	2023	2022	2023	
May	II	10.4	14.7	19.7	25.6	25.2	8.1	Planting
	III	12.0	22.5	23.8	31.6	16.9	0.8	
June	I	17.6	18.5	29.0	28.8	38.0	16.3	Beginning of germination
	II	18.9	17.1	28.9	28.4	7.5	2.8	Full germination
	III	19.3	16.7	29.3	31.3	18.1	25.8	
July	I	19.5	23.8	31.5	37.3	2.7	13.1	Full bloom
	II	22.8	19.5	32.9	34.2	43.4	9.6	Tuber formation
	III	20.8	22.8	26.4	37.3	18.3	12.8	
August	I	23.3	23.8	32.2	34.6	0	0.6	
	II	21.7	24.0	30.4	34.3	0	0.4	
	III	23.0	15.2	35.0	29.9	0	18.4	Harvesting

## Data Availability

All relevant data can be found within the manuscript and its Appendix A.

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
