# Peer review of "Phenotyping of Potato Plants Using Morphological and Physiological Tools"

_plants, 2024, doi:10.3390/plants13050647_

Round 1
Reviewer 1 Report
Comments and Suggestions for Authors
Rozentsvet et al. studied the the phenotyping of potato plants using visualizing tools, which was benefit for predicting key agricultural characteristics. Here are some comments for improving.
1. Some mistakes: line 21, there are two “such”. line 48, there should be a space between Arabidopsis and thaliana. The format of fig2, see the column, they were near by each other. See fig1 and 3, there is a space missing between two columns. Fig5, there is no graphic symbol for different colors. Please use uniform colors throughout the manuscript: fig 6 is in true black and white, while fig 5 is in grey and white? Fig4 is another grey. Fig3 is another kind of grey and black. Fig2 is not the true black.
2. Fig8, morpho-physiological traits of leaves (A-I), however, fig8B is tuber weight, fig 8F is number of stems. Are you sure it belongs to traits of leaves?
3. Table S1, regarding the number of tubers per plant, is the data average numbers of tubers? Otherwise, I believe the number of tubers should be whole values.
4. Fig 6, I suggest labeling virus X, Y, L, S on the y-axis for better understanding. Otherwise, it is difficult to get the information from the figure.
5. Table S2, are the data in table representing days? Also, what is the unit for the data in table S3. Please clarify this information.
6. Line 303, An assessment of genotypic and phenotypic correlations showed that..... (Fig 8H,I). However, we cann’t see any correlation data from fig 8, especially for the data of R values.
7. Conclusion: It needs improvements as well. For example, line 383-385, the correlation between .... was lower .... From the value of R, it is right, the value is lower. However, it should not be considered as a conclusion. What is the meaning of lower value, we can’t predict field virus resistance and yield based solely on this.
Author Response
Dear Reviewer,
We thank the reviewers for their insightful comments that have helped us to prepare a new version.
On detail, we offer the following responses:
- Some mistakes: line 21, there are two “such”. line 48, there should be a space between Arabidopsis and thaliana. The format of fig.2, see the column, they were nearby each other. See fig1 and 3, there is a space missing between two columns. Fig.5, there is no graphic symbol for different colors. Please use uniform colors throughout the manuscript: fig. 6 is in true black and white, while fig. 5 is in grey and white? Fig.4 is another grey. Fig.3 is another kind of grey and black. Fig.2 is not the true black.
Answer: All comments have been taken into account, the figures have been corrected and included in the text and are marked in yellow.
- Fig.8, morpho-physiological traits of leaves (A-I), however, fig. 8 B is tuber weight, fig. 8F is number of stems. Are you sure it belongs to traits of leaves?
Answer: corrected
- Table S1, regarding the number of tubers per plant, is the data average numbers of tubers? Otherwise, I believe the number of tubers should be whole values.
Answer: corrected
- Fig 6, I suggest labeling virus X, Y, L, S on the y-axis for better understanding. Otherwise, it is difficult to get the information from the figure.
Answer: corrected
- Table S2, are the data in table representing days? Also, what is the unit for the data in table S3. Please clarify this information.
Answer: corrected
- Line 303, An assessment of genotypic and phenotypic correlations showed that..... (Fig 8H,I). However, we can’t see any correlation data from fig 8, especially for the data of R values.
Answer: corrected
- Conclusion: It needs improvements as well. For example, line 383-385, the correlation between .... was lower .... From the value of R, it is right, the value is lower. However, it should not be considered as a conclusion. What is the meaning of lower value, we can’t predict field virus resistance and yield based solely on this.
Answer: corrected
Reviewer 2 Report
Comments and Suggestions for Authors
MS is well written and well set out which provides an insight for phenotyping of potatoes through visual morpho-physiological tools. It would be of great intertest for the scientific community.
Author Response
Thank you very much for your positive assessment of our work.
Reviewer 3 Report
Comments and Suggestions for Authors
1. The English writing appears to require some modifications. For instance, "in the second decade of May" is not clear.
2. The introduction mentions many emerging phenotyping techniques, but this study does not seem to have applied these techniques. It is unclear why it is necessary to discuss these relevant references in the introduction and what their relationship is to this study.
3. The title of the manuscript is "Phenotyping of Potato Plants Using Visualizing Morphological and Physiological Tools," but we cannot find corresponding techniques in the Materials and Methods section.
4. "Each type of analysis was carried out in three biological and three analytical replicates" is not clear, and we do not know how many plants were included in each replicate. If only three plants were used, the results may not be representative or statistically significant.
5. The manuscript states that the aim of the work was to investigate the morphometric and morphophysiological parameters of phenotyping potato plants and their relationship with tuber yield. However, Figures 1 to 6 in the results section do not seem to support this objective. It only describes differences in performance among three different ripening periods. The relationship between the results and the research objective is unclear, and it is not clear why this data is presented.
6. Figures 7 to 8, if they are results, should not be placed in the Discussion section. It is an unusual format for a scientific research paper.
7. From the results in Figure 7, we still cannot obtain a key indicator that can be used to predict the ranking of potato varieties for high yield, whether in terms of physiological or morphological data. It can only be inferred that a group of higher-yielding varieties has certain characteristics, but there is no discernible pattern that determines yield.
8. If the author wants to achieve the intended goal, it is necessary to focus on a single variety rather than a large group of varieties. This study does not provide a research report that meets the current requirements for novelty in high-quality journals. It is not recommended to make modifications.
Comments on the Quality of English Language1. The English writing appears to require some modifications. For instance, "in the second decade of May" is not clear.
Author Response
Dear Reviewer,
We thank the reviewers for their insightful comments that have helped us to prepare a new version.
On detail, we offer the following responses:
- The English writing appears to require some modifications. For instance, "in the second decade of May" is not clear.
Answer: corrected
- The introduction mentions many emerging phenotyping techniques, but this study does not seem to have applied these techniques. It is unclear why it is necessary to discuss these relevant references in the introduction and what their relationship is to this study.
Answer: We are grateful for the comment. The above-mentioned references demonstrate our awareness of existing research in this field of study. However, we accept your comment and exclude them from the article. The list of references has been revised accordingly.
- The title of the manuscript is "Phenotyping of Potato Plants Using Visualizing Morphological and Physiological Tools," but we cannot find corresponding techniques in the Materials and Methods section.
Answer: The methodology is outlined and refined. Highlighted in color.
- "Each type of analysis was carried out in three biological and three analytical replicates" is not clear, and we do not know how many plants were included in each replicate. If only three plants were used, the results may not be representative or statistically significant.
Answer: corrected
- The manuscript states that the aim of the work was to investigate the morphometric and morphophysiological parameters of phenotyping potato plants and their relationship with tuber yield. However, Figures 1 to 6 in the results section do not seem to support this objective. It only describes differences in performance among three different ripening periods. The relationship between the results and the research objective is unclear, and it is not clear why this data is presented.
Answer: Figures 1-6 show the differences between potato varieties of different ripeness groups with different morphometric parameters. Next, to identify the main morpho-physiological parameters of phenotypes important for tuber yield, all varieties were divided into three groups (Fig. 7).
- Figures 7 to 8, if they are results, should not be placed in the Discussion section. It is an unusual format for a scientific research paper.
Answer: Figure 7 is moved to the Results
- From the results in Figure 7, we still cannot obtain a key indicator that can be used to predict the ranking of potato varieties for high yield, whether in terms of physiological or morphological data. It can only be inferred that a group of higher-yielding varieties has certain characteristics, but there is no discernible pattern that determines yield.
Answer: Our results indicate that the high yield of varieties is determined by the following phenotypic characteristics: 1) high plant height, large total leaf area; 2) the optimal ratio of bush height to leaf mass area; 3) rapid transition from the stage of planting tubers to the stage of full germination.
- If the author wants to achieve the intended goal, it is necessary to focus on a single variety rather than a large group of varieties. This study does not provide a research report that meets the current requirements for novelty in high-quality journals. It is not recommended to make modifications.
Answer: We suppose that the intended goal can only be achieved by studying as many potato varieties as possible with different morphometric parameters (number of stems, leaf surface area, speed of phenological phases, growth dynamics, etc.). When studying only one variety, it is unlikely to identify the relationship between these parameters and productivity.
Round 2
Reviewer 3 Report
Comments and Suggestions for Authors
1. While the author provided methods for image capture, there is no explanation of how relevant features are extracted, making it unclear how the images are translated into information about various morphological characteristics. Factors like height and angle during capture can affect the final data extraction. Furthermore, the study is titled "Phenotyping of Potato Plants Using Visualizing Morphological and Physiological Tools," but it appears that very few features are obtained using visual tools, and there is no information on which features were obtained using visual tools.
2. Conducting trait selection with only three replicates is too limited.
3. It is unclear why sometimes n is set to 3 and other times n is set to 9.
4. The method mentions using 3 years of weather data to assess its impact on crop growth, but the results don't explain how these weather changes affected the crop growth. If the conclusion is merely that weather conditions impact growth, this isn't a novel scientific argument but a widely known phenomenon.
5. The statement "This confirms that a combination of morphological and physiological parameters determines the differentiation of phenotypes by yield" was added by the author, but we can't derive such conclusions from the provided information. We still can't determine which morphological features are critical, their relative importance, or their significance from Figure 7. In other words, we still can't discern which morphological features can serve as key breeding indicators from the current information.
6. The statement "Thus, growing conditions and genotypic characteristics have a significant impact on the growth traits of potato varieties" cannot be inferred from the data provided by the author. The article presents some growth data, but the connection between genotype and growth traits is unclear. It's a widely known fact that different genes affect crop growth differently, and the significance of explicitly mentioning this is unclear.
7. Figure 5 compares data from different years and then discusses differences in maturity levels. Such comparisons blend annual variations with maturity levels in statistical analysis, which can lead to ambiguous conclusions. The data in Figure 5 shows that early and middle-early varieties do not always have higher LMA, and many related features do not match the author's descriptions, especially when based on flawed statistical analysis.
8. Regarding the earlier question about the study's focus, the author responded, "We suppose that the intended goal can only be achieved by studying as many potato varieties as possible...," indicating agreement with the idea of studying individual varieties to obtain comprehensive information. However, the majority of the discussion groups varieties into "early," "middle," and "late" maturity categories rather than comparing individual varieties. If it's not feasible to compare features for each individual variety and clustering is necessary, it suggests that the features identified so far may not be applicable to every variety and need normalization and standardization across multiple varieties. In other words, using these techniques for breeding may only yield a group of varieties rather than selecting the best individual variety.
9. Although the author provided some responses and made "minimal" changes to the manuscript, it still doesn't address issues related to novelty and statistical analysis. Incorrect statistical analysis makes it challenging to draw accurate conclusions.
10. This manuscript still does not meet the standards for publication in a Q1 journal. Due to insufficient novelty and incomplete or incorrect statistical analysis, it is challenging for us to endorse such a manuscript.
Comments on the Quality of English LanguageThe author did not make revisions to the previous manuscript in English
Author Response
Dear reviewer!
Once again, we thank you for your thorough analysis of our article and constructive comments. We have critically revised our results in a new version of the article.
Title corrected: Phenotyping of Potato Plants Using Morphological and Physiological Tools
Other our notes are explained below:
Rew.: While the author provided methods for image capture, there is no explanation of how relevant features are extracted, making it unclear how the images are translated into information about various morphological characteristics. Factors like height and angle during capture can affect the final data extraction. Furthermore, the study is titled "Phenotyping of Potato Plants Using Visualizing Morphological and Physiological Tools," but it appears that very few features are obtained using visual tools, and there is no information on which features were obtained using visual tools.
Answer: The visually measured parameters were the number of tubers, the height of the plants, the number of stems in the bush, the width and length of the lateral lobes of the leaf, the area of the leaf mass of the bush, the duration of the period from the beginning of planting to the appearance of seedlings, the duration of the period between complete germination to the beginning of flowering, and the duration of flowering, appearance of leaves affected by infection.The study was carried out on 29 potato varieties. Each variety was planted with 50 tubers in four replicates, i.e. 200 tubers. Not all tubers sprouted, so the sample size varied depending on the variety, namely, leaves from 15-20 plants of each variety were used for further measurements. The height and growth dynamics of plants (10-15 bushes) of each variety were determined randomly.To work with the phenotype of plants of 29 varieties, a photography method was used using a digital camera. Next, the images were processed in the JMicroVision program. To determine the bush area of the plant, each variety, photographs were taken from above and from the side at right angles. A total of 320 photographs were taken with the goal of minimizing errors that could arise during the work. Separately, photographs were taken of large portions of plant leaves from 15-20 plants of 3-4 leaves each, which were further processed in the JMicroVision program.
Figure. Examples of obtaining digital images for phenotyping.
Rew.:. Conducting trait selection with only three replicates is too limited.
Answer:
Data in tables are presented as absolute values: as average values for a group of varieties in bar graphs, as average values of measured parameters for each variety in scatter plots
Specific leaf area (SLA) was determined by obtaining cuttings from the middle part of 5-10 leaves for each variety which were then weighed, dried to a constant weight and used for calculation.
The specific LMA was determined by the formula m/S (mg/dm2), where m– the amount of dry mass (mg), S– the leaf surface area (dm2).
Stomata were counted on longitudinal paradermal sections of leaves, previously fixed in 3.5% glutaraldehyde (pH = 7.5). The number of stomata per 1 cm2 on the lower surface of the leaf was counted, their length was measured and expressed in µm. A biological microscope “OPTIKA” B-500TPL (Italy) was used for analysis
The lateral lobes of leaves were collected from 15-20 plants of each variety to measure physiological parameters. They were cut and three biological samples of 0.1-0.5 g each were taken depending on the type of analysis.Each biological sample was analyzed three times.
Rew.:. It is unclear why sometimes n is set to 3 and other times n is set to 9.
Answer: explanation above. For example, ER – early ripening (n = 9), ME – middle-early (n = 9), MSR – mid–season ripening (n = 11). Each bar represents the mean ± SE (n – number of varieties) or .HYV – high-yielding varieties (n = 8), SV – standards and varieties equal to standards (n = 9), LYV – low-yielding varieties (n = 12) (n – number of varieties);
Rew.:.. The method mentions using 3 years of weather data to assess its impact on crop growth, but the results don't explain how these weather changes affected the crop growth. If the conclusion is merely that weather conditions impact growth, this isn't a novel scientific argument but a widely known phenomenon.
Answer: To exclude annual fluctuations in the measured traits, average values were calculated for two years.
The goal of the work has been adjusted: The aim of the research was to study the phenotypic variability of potato genotypes using visual morphological and physiological traits associated with yield. The tasks of the study were: (i) assessment of the yield of 29 new potato varieties in different varietal groups grown in the climatic conditions of the Middle Volga region of Russia in 2022-2023; (ii) study of growth dynamics according to phenological indicators; (iii) test of resistance to viral diseases.
Rew.:The statement "This confirms that a combination of morphological and physiological parameters determines the differentiation of phenotypes by yield" was added by the author, but we can't derive such conclusions from the provided information. We still can't determine which morphological features are critical, their relative importance, or their significance from Figure 7. In other words, we still can't discern which morphological features can serve as key breeding indicators from the current information.
The statement "Thus, growing conditions and genotypic characteristics have a significant impact on the growth traits of potato varieties" cannot be inferred from the data provided by the author. The article presents some growth data, but the connection between genotype and growth traits is unclear. It's a widely known fact that different genes affect crop growth differently, and the significance of explicitly mentioning this is unclear.
Answer: We conducted additional statistical analysis using scatter plots and correlations. The conclusions have been corrected.
Rew.: Figure 5 compares data from different years and then discusses differences in maturity levels. Such comparisons blend annual variations with maturity levels in statistical analysis, which can lead to ambiguous conclusions. The data in Figure 5 shows that early and middle-early varieties do not always have higher LMA, and many related features do not match the author's descriptions, especially when based on flawed statistical analysis.
Answer: corrected
Rew.: Regarding the earlier question about the study's focus, the author responded, "We suppose that the intended goal can only be achieved by studying as many potato varieties as possible...," indicating agreement with the idea of studying individual varieties to obtain comprehensive information. However, the majority of the discussion groups varieties into "early," "middle," and "late" maturity categories rather than comparing individual varieties. If it's not feasible to compare features for each individual variety and clustering is necessary, it suggests that the features identified so far may not be applicable to every variety and need normalization and standardization across multiple varieties. In other words, using these techniques for breeding may only yield a group of varieties rather than selecting the best individual variety.
Answer: Additional statistical analysis was performed. Added fig. 9.
Rew.: Although the author provided some responses and made "minimal" changes to the manuscript, it still doesn't address issues related to novelty and statistical analysis. Incorrect statistical analysis makes it challenging to draw accurate conclusions.
Answer: We regret that we were not able to immediately clearly present the novelty of our work.
We hope that in the new version we have answered the questions raised.
Rew.: This manuscript still does not meet the standards for publication in a Q1 journal. Due to insufficient novelty and incomplete or incorrect statistical analysis, it is challenging for us to endorse such a manuscript.
Answer: An analysis of a publication in the journal Plants devoted to the problem of phenotyping shows that our study meets modern trends in this area, allows us to understand the importance of phenotypic variations and the use of morphological traits associated with yield of potato.
Round 3
Reviewer 3 Report
Comments and Suggestions for Authors
The author has already made many optimizations, but the experimental design clearly shows that this study is not well-designed, primarily because the varying number of varieties in each category leads to disparities in assessment. Additionally, having only 3 repetitions also seems quite inappropriate. However, if the editor feels that such research results meet the journal's requirements, then there are no further comments.
Author Response
Dear Reviewer,
There are no comments in the latest review that could be corrected. As for planning the work, this is the prerogative of the authors. When working with a large number of potato varieties, as well as other crops, it was necessary first of all to determine their classification. In this case, attribute them to the ripening period, which is what we did. Then we divided all varieties into groups according to yield. Only then were we able to identify signs that contribute to improved performance. From our point of view, this plan is logical. The number of repetitions and the number of varieties in each group do not contradict statistical principles. We insist that the work was performed at a high professional level and meets the requirements of Q1 magazine.